# PlugMed: Improving Specificity in Patient-Centered Medical Dialogue Generation using In-Context Learning

**Chengfeng Dou, Zhi Jin✉, Wenpin Jiao✉, Haiyan Zhao,**
**Yongqiang Zhao**, **Zhenwei Tao**
School of Computer Science, Peking University;
Key Laboratory of High Confidence Software Technologies(PKU), MOE, China
{chengfengdou,zhijin,jwp,zhhy.sei}@pku.edu.cn
{tttzw,yongqiangzhao}@stu.pku.edu.cn

## Abstract

The patient-centered medical dialogue systems strive to offer diagnostic interpretation services to users who are less knowledgeable about medical knowledge, through emphasizing the importance of providing responses specific to the patients. It is difficult for the large language models (LLMs) to guarantee the specificity of responses in spite of its promising performance even in some tasks in medical field. Inspired by in-context learning, we propose PlugMed, a Plug-and-Play Medical Dialogue System, for addressing the challenge. PlugMed is equipped with a prompt generation (PG) module and a response ranking (RR) module to enhances LLMs' dialogue strategies for improving the specificity of the responses. The PG module is used to stimulate the imitative ability of LLMs by providing them with real dialogues from similar patients as prompts. The RR module incorporates fine-tuned small model as response filter to enable the selection of appropriate responses generated by LLMs. Furthermore, we introduce a new evaluation method based on matching both user's intent and high-frequency medical term to effectively assess the specificity of the responses. We conduct experimental evaluations on three medical dialogue datasets, and the results, including both automatic and human evaluation, demonstrate the effectiveness of our approach.

## 1 Introduction

As a key task in health conversational assistants, Medical Dialogue Generation aims to automatically generate informative responses to the users. It's better for such dialogues to be medical knowledgeable, patient-specific and context-aware, as the patients may be less knowledgeable about medical knowledge and their dialogue contexts may vary.

Prior studies (Li et al., 2021a; Varshney et al., 2023b,a) have emphasized the critical role of domain knowledge in medical dialogue generation,

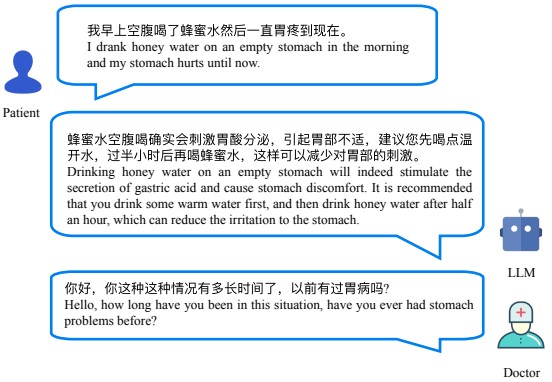

我早上空腹喝了蜂蜜水然后一直胃疼到现在。
I drank honey water on an empty stomach in the morning and my stomach hurts until now.

Patient

蜂蜜水空腹喝确实会刺激胃酸分泌，引起胃部不适，建议您先喝点温开水，过半小时后再喝蜂蜜水，这样可以减少对胃部的刺激。
Drinking honey water on an empty stomach will indeed stimulate the secretion of gastric acid and cause stomach discomfort. It is recommended that you drink some warm water first, and then drink honey water after half an hour, which can reduce the irritation to the stomach.

LLM

你好，你这种这种情况有多长时间了，以前有过胃病吗？
Hello, how long have you been in this situation, have you ever had stomach problems before?

Doctor

Figure 1: An example of the medical dialogue. The LLM and the doctor adopt different response strategies for the same question.

many of which integrate general medical knowledge from knowledge bases into the models by utilizing knowledge injection mechanism. As the scale of large language models (LLMs) continues to expand, the lack of knowledge is being alleviated. This is evident in the performance of LLMs such as Instruct GPT (Ouyang et al., 2022), which outperform small models based on fine-tuning (Singhal et al., 2022) on medical question answering tasks without any additional training.

However, LLMs have been criticized for lacking medical diagnostic logic to the patient, which can lead to untargeted and even risky response suggestions (Howard et al., 2023; Zhang et al., 2023). For instance, as illustrated in Figure (1), when a patient claims to have a particular ailment and seeks medication, the doctor's priority is to first investigate the underlying disease to make targeted suggestions. In contrast, LLMs are more inclined to give straightforward medical advice, rather than further gathering patient information to give accurate advice. Once LLMs give overconfident advice, it is difficult for patients to discern the effectiveness and safety of such advice.

Researches have demonstrated that in-context

learning (ICL) possesses the ability to impact the LLMs' conversational style and mitigate prejudice and toxicity concerns (Roy et al., 2023; Meade et al., 2023) by demonstrating a few examples. This phenomenon is due to the powerful imitation learning and few-shot capabilities of LLMs, by learning patterns from a small number of samples and applying them in generation (Dong et al., 2022).

Taking inspiration from these studies, we propose leveraging ICL to shape the LLMs' dialogue strategies and accordingly design a Plug-and-Play Medical Dialogue System, named PlugMed, which embodies two crucial components: a Prompt Generation (PG) Module and a Response Ranking (RR) Module. Specifically, PlugMed uses the PG module to identify examples by considering information from both global and local views. From global view, the PG module choose relevant examples for ensuring that the model acquires a comprehensive understanding of the entire dialogue process by exploiting the similarity with the entire dialogue history. Conversely, from local view, the PG module priorities recent utterances to capture the most relevant information for generating responses. To further maximize advantages of both the global and local views, PlugMed uses the RR module to autonomously select the most appropriate response for the ongoing dialogue through utilizing a fine-tuned small model.

Another critical consideration is about appropriate automatic evaluation metrics for medical dialogue systems. Previous studies (Varshney et al., 2023b; Zhang et al., 2023) only relied on open-domain dialogue evaluation methods. However, as indicated in (Ji et al., 2023), these evaluation methods may be unreliable in task-oriented scenarios. To gain a comprehensive understanding of the system's real-world performance, we undertake a thorough evaluation that is twofold: the intent accuracy and the high-frequency medical term accuracy. Here, the intent accuracy is used to evaluate the reasonableness of the dialogue actions adopted by the system, and the high-frequency medical term accuracy focuses on measuring the presence of essential medical information in the system's responses.

We evaluate our approach on three widely used large medical datasets, i.e., Meddg, MedDialogue and Kamed datasets. Both automatic and human evaluations show that our approach can substantially improve the specificity of LLMs. Our contributions can be summarized as follows:

- An ICL-based approach that enhances LLMs to generate responses that conform to the diagnostic strategy.

- The comprehensive evaluation metrics for medical dialogue automation that take both the intent accuracy and the high-frequency medical term accuracy into account.

- Experiments demonstrating the key elements of automated medical diagnosis.

## 2 Methodology

### 2.1 Overview

We propose a framework as shown in Figure (2), which employs ICL to guide the LLM towards generating high-quality replies, where the Prompt Generation (PG) Module and the Response Ranking (RR) Module are two key components. The PG Module accepts the dialogue history as input and outputs multiple In-context prompts, along with a single Instruct prompt. Based on these prompts, the LLM generate multiple system responses. Then, the RR Module utilizes a small language model (SLM) to select the best response. We will elaborate these two components in the following subsections.

### 2.2 Prompt Generation Module

The PG module uses a multi-strategy retrieval framework to retrieve examples similar to the input sample and then employ them to generate prompts.

#### 2.2.1 Basic Ideas

As shown in 'Dialogue History' at the left part of Figure (2), the main idea is as follows. Firstly, from the global view, we considers the entire history of the dialogue, and the global retriever retrieves the similar dialogues from the the training set of the dataset. Secondly, considering the global view is susceptible to distractions caused by the abundance of irrelevant information, which may lead to inappropriate retrieved examples, we include the local view for enhancing the relevance of the retrieval. Concretely, the local retriever first extracts the patient's symptom information from past conversations, serving as the initial filter for the samples. Considering the recent rounds of conversations hold the utmost relevance to the system-generated responses, we then utilize these conversations as query to select examples. This effectively mitigate the interference of irrelevant information.

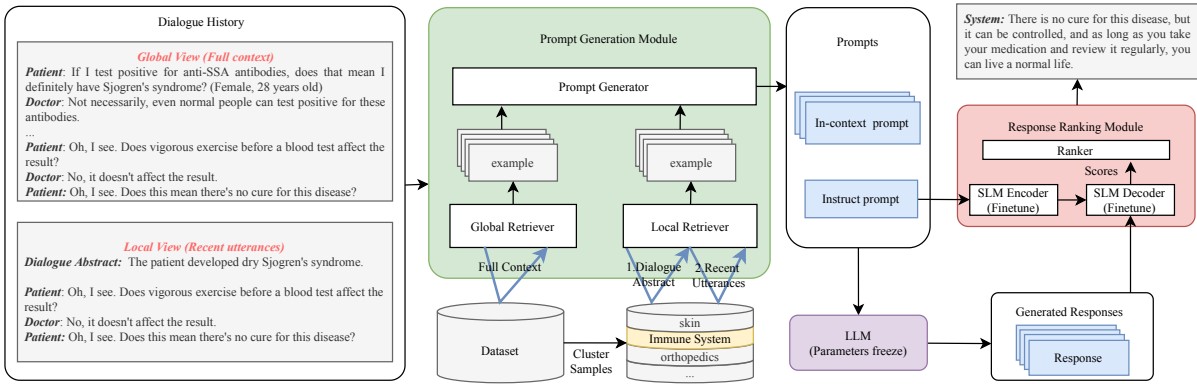

Figure 2: The overview of PlugMed. Our system consists of two core components, i.e., *Prompt Generation (PG) Module*, which retrieves similar examples in the dialogue history from both global and local views to generate prompts, and *Response Ranking (RR) Module*, which ranks the outputs of LLM corresponding to these prompts and selects the best responses.

Thirdly, we employ the retrieved examples from both views to take advantage of their respective strengths.

### 2.2.2 Implementation

**Global Retriever.** The global retriever utilizes the full context of dialogue history as a query for searching samples. It employs Sentence-Bert (SBERT) (Reimers and Gurevych, 2019) to encode the query and examples, and then utilizes cosine similarity to identify the closest examples.

**Local Retriever.** The local retriever retrieves samples in terms of symptoms and recent utterances. For getting symptoms, we develop a medical dialogue summary model that utilizes the ICMS-MRG (Chen et al., 2022) dataset in conjunction with BART (Lewis et al., 2019) as the backbone model. This model enables the extraction of the chief complaint, containing the patient symptom information, and we use SBERT as the encoder of chief complaints to provide embeddings for the following operations. We first encode the chief complaints of examples and use the K-Means algorithm to divide them into $\mathcal{K}$ groups. Then, we extract the embeddings of each cluster centers to serve as the symptom index for querying. When performing the search, we first retrieve the candidate examples by computing the cosine similarity between the embeddings of the sample's chief complaint and the symptom index. Then, we use SBERT to retrieve examples from the candidates based on recent dialogue utterances.

**Prompt Generator.** The prompt generator generates two types of prompts, as depicted in Figure (2):

'Instruct prompt' and 'In-context prompts'. Each In-context prompt corresponds to a distinct example retrieval strategy.

It is needed to compress examples to include more demonstrations, given the input length constraint imposed by the LLM, when generating the in-context prompts. Drawing inspiration from Hu et al. (2022), for each example, we keep only the most recent rounds of conversations, and we restrict the maximum conversation length to no more than $n$. Moreover, we employ the previous mentioned chief complaints (up to $m$ characters long) as the dialogue abstract, replacing the excluded history to achieve dialogue compression.

The Instruct prompt includes the full context, and we include the instruction before the history to prompt LLM to act as a doctor. We use this prompt to activate the zero-shot capability of LLMs. Appendix E gives some examples of both prompts.

### 2.3 Response Ranking Module

Through the experimentation, we observe that LLMs tends to generate a significant number of medical terms during dialogue, resulting in more comprehensive responses. However, LLMs often exhibit overconfidence, as they question patients less frequently, resulting in a decrease in the overall quality of responses. On the contrary, small language models (SLMs) that are fine-tuned on a dialogue corpus behave cautiously and tend to include a limited number of medical terms in their output. This phenomenon is illustrated in Figure (3). Acknowledging the complementary nature of LLMs and SLMs, we propose using a SLM to evaluate the responses of the LLM.

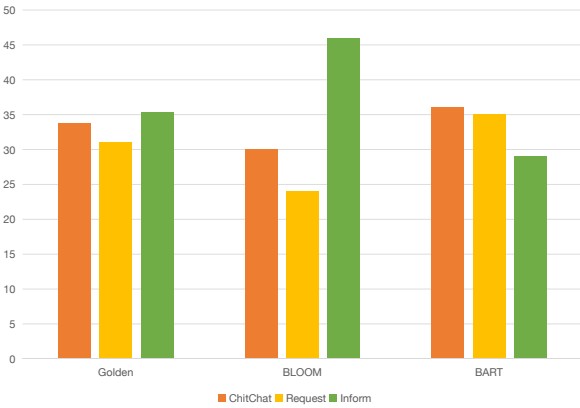

Figure 3: We compared the behavior patterns of LLMs and SLMs, using Bloom (Scao et al., 2022) as a representative of LLMs and Bart (Lewis et al., 2019) as a representative of SLMs. We counted the distribution of dialogue actions taken by them on the validation set of KaMed (Li et al., 2021b), where the human action is denoted as Golden. The *Request* action indicates the collection of patient information, while the *Inform* action indicates the provision of advice to the patient.

Specifically, for a given sample with a dialogue history $h$, we use perplexity as the score of the response $r$ generated by the LLM. We compute the score using the following equation:

$$s(r) = -\frac{1}{l} \sum_{i=1}^{l} \log p(r_i \mid r_{<i}, h; \theta) \qquad (1)$$

Here, $l$ represents the length of response $r$, and $\theta$ denotes the model parameter of the SLM. This evaluation model uses an encoder-decoder architecture, where $h$ is input to the encoder side of the model, and $r_{<i}$ is input to the decoder side for calculating the generation probabilities. We select the response with the lowest score as the system output.

## 3 Automatic Evaluation Metrics

We observe that previous studies (Varshney et al., 2023b,a; Zeng et al., 2020) employ the metrics for open domain dialogue tasks that often cannot effectively measure system performance in task-oriented settings (Ji et al., 2023; Risch et al., 2021). From the results given in Figure (3), we observe that the LLM exhibits overconfidence. To judge whether our approach alleviates the problem, it becomes crucial to evaluate the dialogue actions taken by the LLM. It is equally important to evaluate the dialogue content generated by LLM. Hence, we introduce two metrics that consider both intent and the usage of high-frequency medical terms.

### 3.1 Intent Evaluation

The intent accuracy (Int) is used to assess the conformity of the system's response to the ground-truth in terms of dialogue actions. We train a medical dialogue intent classifier for calculating Int, and Int is calculated by the following formula:

$$Int = \frac{1}{N} \sum_{i=1}^{N} f(Pred_i, Golden_i) \qquad (2)$$

Here, $N$ represents the total number of samples. $Pred_i$ and $Golden_i$ denote the model's predicted response and the corresponding actual response, respectively. The function $f(\cdot, \cdot)$ evaluates the intentions, which are extracted by the aforementioned intent classifier, assigning a value of 1 if the intentions of the two responses are the same, and a value of 0 otherwise. Appendix A.1 presents the implementation details of the intent classifier.

### 3.2 Medical Term Evaluation

We evaluate the completeness and correctness of the responses using the micro-f1 score, which measures the overlap of high-frequency medical terms between the ground-truth and the prediction. We need to avoid relying solely on exact matching when measuring the term overlap. For instance, different doctors may prescribe different medications for some diseases that may have same effect on the patient. Employing exact matching alone may result in an underestimation of the performance exhibited by models that possess the ability to generate diverse treatment options.

Hence, we introduce a novel approach called Top-n Match (TnM) to address the problem. Concretely, let $T = \{t_1, t_2, ..., t_{|T|}\}$ be a set consisting of $|T|$ terms, and let $s$ be a similarity score function that satisfies $0 \leq s(t_i, t_j) \leq 1$ for all $t_i, t_j \in T$. For a given term $t_i \in T$, we define $S_i^n(s, t_i) \subseteq T$ as the set of $n$ terms that are closest to $t_i$ based on the similarity function $s$. We say that $t_i$ and $t_j$ are Top-N Match if and only if $S_i^n(s, t_i) \cap S_j^n(s, t_j) \neq \emptyset$. TnM with different $n$ can represent the term-matching scores for different similarity levels. We present the f1-score results using T3M configurations. Appendix A.2 and A.3 will give more details about the term extraction and matching.

## 4 Experiments

This section delineates the evaluation setup and the results of the proposed approach in the context of

| Dataset | Train / Valid / Test | Turn |
|---------|---------------------|------|
| Meddg | 14,864 / 2,000 / 1,000 | 9.92 |
| MedDialog | 32,723 / 3,000 / 3,000 | 4.76 |
| KaMed | 57,754 / 3,000 / 3,000 | 11.62 |

Table 1: The statistics of datasets, where *Turn* indicates the average number of rounds contained in each dialogue session contains.

medical dialogue generation.

## 4.1 Datasets and Evaluation Metrics

We conduct experiments on three large datasets for our evaluation. 1) The **Meddg** dataset (Liu et al., 2022) which is collected from *Doctor Chunyu*[1] and consists of 17,864 dialogue sessions. 2) The **MedDialogue-CN** dataset (Zeng et al., 2020) which is collected from *HaoDaifu*[2] and comprises 38,723 dialogues without any provided annotations. 3) The **KaMed** dataset (Li et al., 2021b) which is also collected from *Doctor Chunyu*, but at a larger scale, containing 63,754 dialogue sessions. Statistics of three datasets are presented in Table (1).

To evaluate the quality of generation, we employ the Rouge-L (Lin, 2004) and Bert-Score (Zhang et al., 2019) to measure the overall similarity between the generated text and ground-truth. We also utilize micro-F1 scores for entity matching in T3M settings to assess medical term correctness based on the aforementioned definitions. Additionally, we employ the INT metric to measure the accuracy of the intended responses. The validity analysis of each metric can be found in the Section 4.8.

## 4.2 Implementations

Our approach employs BLOOM as the foundation model, and to ensure reproducibility, we avoid any form of sampling and instead utilize a greedy decoding strategy. We generate a set of four prompts for a given sample, including an Instruction prompt (referred to as "Vanilla") and three In-context prompts. These prompts are as follows: 1) **Vanilla**: This strategy instructs the model to act as a doctor by prefixing the samples with an instruction. 2) **Global View**: This strategy involves looking for examples through a global retriever, using full context as query. 3) **Local Primary**: In this strategy, we first consider the patient's chief

complaint to retrieve examples with similar symptoms. From these examples, a number of samples are randomly selected to generate the prompt. This strategy corresponds to the initial step of the local retriever's two-step search. 4) **Local Secondary**: This strategy searches for examples through a local retriever and utilizes a full two-step search. All examples are from the training set of the dataset.

Limited by input length, the In-context prompt contains 4 examples, while each example is limited to the last 140 ($m = 20, n = 120$) tokens. When constructing the symptom index using the K-Means algorithm, we set the number of cluster centers to 100 and iterate 20 times. During response ranking, BART serves as the scoring model. We conduct the experiments using PyTorch[3] and Huggingface Inference API[4].

## 4.3 Baselines

**Fine-tuning Based Baselines.** These baselines utilize small language models as the backbone, trained on the aforementioned datasets for the medical dialogue generation task, which include the following models: 1) **Bart** (Lewis et al., 2019), a well-known encoder-decoder model that is recognized for its text generation capabilities. 2) **Mars** (Sun et al., 2022), an advanced model explicitly crafted for the MultiWoZ (Budzianowski et al., 2018) dataset, renowned for its exceptional performance in addressing the Task-Oriented Dialogue (TOD) task. We migrated Mars into our tasks and trained it to focus on the generation of medical terms. Appendix B gives the further information on the migration process.

**LLM Baselines.** These baselines employ large language models to generate dialogue responses. Our comparison targets include: 1) **Bloom** (Scao et al., 2022), a widely used open-source multilingual language model with 176 billion parameters. 2) **Bloomz** (Scao et al., 2022), an instruction-tuned model derived from Bloom and specializes in zero-shot tasks. In addition to these models that use Instruct-prompt as input, we also include two baselines that utilize In-context prompts as input: 1) **ICL Rand**, which selects a set of dialogue examples randomly to construct the prompt. 2) **ICL Sbert**, which utilizes Sbert (Reimers and Gurevych, 2019) to encode the dialogue history and find the

---

[1] https://www.chunyuyisheng.com/
[2] https://haodf.com
[3] https://pytorch.org/
[4] https://huggingface.co/docs/api-inference/index

| Model | Meddg | | | | MedDialogue | | | | KaMed | | | |
|---|---|---|---|---|---|---|---|---|---|---|---|---|
| | R-L | B-S | T3M | INT | R-L | B-S | T3M | INT | R-L | B-S | T3M | INT |
| *Fine-tuning based* | | | | | | | | | | | | |
| Bart | 24.1 | 64.8 | 9.9 | 42.0 | 11.8 | 59.0 | 16.2 | 42.0 | 15.9 | 60.1 | 15.5 | 45.0 |
| Mars | 20.9 | 63.1 | 10.3 | 39.9 | 10.5 | 58.4 | 16.8 | 37.7 | 12.7 | 59.3 | 16.4 | 41.4 |
| *LLM based* | | | | | | | | | | | | |
| Bloom | 14.2 | 60.3 | 10.9 | 26.3 | 10.7 | 57.2 | 16.2 | 33.0 | 11.3 | 58.7 | 16.9 | 34.1 |
| Bloomz | 14.5 | 58.9 | 8.5 | 24.8 | 10.0 | 57.9 | 12.4 | 31.9 | 9.7 | 57.8 | 15.0 | 32.0 |
| ICL Rand | 16.4 | 61.2 | 10.6 | 29.6 | 9.9 | 57.3 | 14.8 | 36.6 | 9.3 | 56.9 | 15.0 | 36.1 |
| ICL Sbert | 18.7 | 62.7 | 11.3 | 36.3 | 11.9 | 60.0 | 17.7 | 37.3 | 12.0 | 59.5 | 16.8 | 37.9 |
| PlugMed(ours) | **21.1** | **64.1** | **12.1** | **41.3** | **12.8** | **60.4** | **18.4** | **37.9** | **14.1** | **60.1** | **18.1** | **41.7** |

Table 2: Automatic evaluation on the Meddg, MedDialog and KaMed datasets. R-L and B-S denote Rouge-L and Bert-Score, respectively. Boldface scores indicate best results. The performance improvement of PlugMed over the baselines is significant with $p < 0.05$.

closest examples to the given sample. Both baselines use Bloom as the fundamental model.

## 4.4 Overall Performance

Table (2) present the automatic evaluation results on Meddg, MedDialog, and KaMed. Remarkably, PlugMed consistently attains the top-ranking positions across a majority of the metrics and achieves best performance in T3M, outperforming even the strongest baseline. Meanwhile, among all baselines leveraging the LLM, PlugMed generates responses with the most reasonable intent.

Our analysis uncovers interesting observations. Firstly, we discovered that fine-tuned small models tend to have higher INT scores but lower term matching-related scores. This suggests that while these models excel in emulating the dialogue actions of doctors, they often struggle to generate appropriate medical terminology due to their limited medical knowledge. Secondly, we observed that Bloomz performed worse than Bloom, indicating that the instruct-tuning process may compromise the diagnostic capability of the model. More details can be found in Section 4.6.

## 4.5 Ablation Study

To investigate the influence of different prompt generation strategies on system responses and the efficacy of the RR module, we conduct ablation experiments. The experimental results, shown in Table (3), indicate that both the global and local view effectively enhance the quality of the LLM output. Moreover, we observe a substantial enhancement in the quality of system responses due to the integration of the RR module. This observation underscores the efficacy of the fine-tuned SLM in effectively evaluating the LLM's output.

Furthermore, we observe that Global View demonstrates comparable performance to PlugMed when assessed using the MedDialogue dataset. However, PlugMed significantly surpasses Global View on the Meddg and KaMed datasets. We attribute this discrepancy to the MedDialogue dataset's characteristic of having shorter dialog histories, averaging only 4.76 rounds. Conversely, Global View's retrieval effectiveness diminishes as the samples' length increases, as elaborated in Section 2.2.1, making it less effective than PlugMed on the other two datasets. This underscores the synergistic relationship between Global View and Local View, highlighting their complementary strengths.

To delve deeper into the contributions of these strategies to the outcomes, we utilize the RR module to rank the responses corresponding to each strategy, documenting the percentage of times each strategy achieves the top rank. The results are illustrated in Figure (4). Based on the findings depicted in the figure, we can infer that the responses generated by the Vanilla strategy exhibit significantly inferior quality compared to those produced by the other three strategies. Furthermore, the probability of being selected by the remaining three strategies is approximately equal, indicating their complementary nature.

| Dataset | Settings | Bloom | | | | ChatGPT | | | |
|---|---|---|---|---|---|---|---|---|---|
| | | R-L | B-S | T3M | INT | R-L | B-S | T3M | INT |
| Meddg | Vanilla | 14.2 | 60.3 | 10.9 | 26.3 | 6.7 | 56.1 | 10.5 | 23.4 |
| | Global View | 18.7 | 62.7 | 11.3 | 36.3 | 12.3 | 60.0 | 12.7 | 33.8 |
| | Local Primary | 18.7 | 62.5 | 10.7 | 34.6 | 11.6 | 56.1 | 12.3 | 30.7 |
| | Local Secondary | 18.0 | 62.3 | 11.2 | 34.9 | 11.7 | 59.7 | 12.1 | 30.3 |
| | +Ranking (Ours) | 21.1 | 64.1 | 12.1 | 41.3 | 12.9 | 60.4 | 12.9 | 34.4 |
| MedDialogue | Vanilla | 10.7 | 57.2 | 16.2 | 33.0 | 7.5 | 56.5 | 16.5 | 40.5 |
| | Global View | 11.9 | 60.0 | 17.7 | 37.3 | 10.1 | 58.6 | 17.8 | 38.8 |
| | Local Primary | 11.0 | 59.0 | 15.5 | 36.6 | 10.0 | 58.5 | 18.3 | 38.2 |
| | Local Secondary | 12.0 | 60.0 | 16.7 | 36.1 | 9.9 | 58.4 | 18.0 | 39.1 |
| | +Ranking (Ours) | 12.8 | 60.4 | 18.4 | 37.9 | 10.1 | 58.6 | 18.5 | 40.2 |
| KaMed | Vanilla | 11.3 | 58.7 | 16.9 | 34.1 | 6.1 | 54.9 | 14.9 | 36.2 |
| | Global View | 12.0 | 59.5 | 16.8 | 37.9 | 8.9 | 57.5 | 17.1 | 39.8 |
| | Local Primary | 12.0 | 59.5 | 17.1 | 39.4 | 8.6 | 57.2 | 17.0 | 38.9 |
| | Local Secondary | 12.1 | 59.4 | 17.7 | 36.7 | 8.4 | 57.2 | 17.1 | 38.7 |
| | +Ranking (Ours) | 14.1 | 60.1 | 18.1 | 41.7 | 9.1 | 57.6 | 17.1 | 40.5 |

Table 3: Ablation studies on the Meddg, MedDialog and KaMed datasets. *Vanilla*, *Global view*, *Local primary* and *Local secondary* represent the four prompt generation strategies respectively, and *+Ranking* represents the results of the selection based on the four strategies using the RR module.

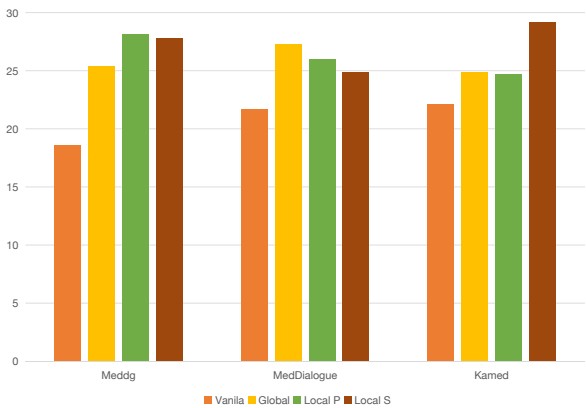

Figure 4: The evaluation results of the RR module for four strategies. The vertical axis represents the proportion of the strategy get the best score. *Local P* and *Local S* correspond to Local Primary and Local Secondary, respectively.

### 4.6 Analysis of Generalization Capacity

We conducted experiments using ChatGPT and Bloomz to assess the effectiveness of our approach in terms of generalization. The results, as presented in Table (3) for ChatGPT, revealed that our approach successfully enhances ChatGPT's performance in completing medical conversations. However, it was observed that ChatGPT's capability for multi-round conversations lags behind that of Bloom. This discrepancy can be attributed to the safety protocols integrated into ChatGPT by OpenAI. Specifically, ChatGPT often advises patients to seek professional assistance instead of providing diagnoses, which affects its multi-round conversation potential. Conversely, we noticed that our approach has minimal impact on Bloomz, as detailed in Appendix C. This disparity can be attributed to Bloomz's training dataset, bigscience/xP3 [5], which primarily comprises single-round Q&A tasks, making it less adaptable to multi-round conversations. To summarize, the generalization ability of our method is influenced by the model's pre-training task, and improving our method's effectiveness entails pre-training models based on multi-round dialogues and reducing safety interventions.

### 4.7 Human Evaluation

We manually evaluate seven selected dialogue models to conduct a comprehensive comparison. We ensure a thorough analysis by randomly selecting 100 samples from each dataset. Each sample is examined by three physicians who assign scores based on the criteria outlined in Table (4). The evaluators check the responses in the following order: Role Consistency, Empathy, Correctness, Neces-

---

[5] https://huggingface.co/datasets/bigscience/xP3

| Check Item | Content |
|---|---|
| Role Consistency | The response should exhibits a doctor-like response style and utilizes natural language. |
| Empathy | The response should demonstrate an understanding of the user's intents and employ a friendly tone. |
| Correctness | The responses must adhere to common medical sense and be evidence-based. |
| Necessity | The response should provide assistance in advancing the diagnosis or satisfying the patient's curiosity. |
| Richness | The responses should incorporate additional relevant medical information to significantly enhance the patient experience. |

Table 4: Check items for human evaluation.

| Model | Human | BLEU | R-L | B-S | T1M | T3M | T5M | INT |
|---|---|---|---|---|---|---|---|---|
| Golden | 90.4 | - | - | - | - | - | - | - |
| Bart | 58.9 | 4.1 | 22.1 | 63.0 | 8.3 | 11.6 | 12.3 | 40.9 |
| Mars | 44.6 | 5.4 | 19.2 | 62.3 | 9.1 | 12.8 | 13.9 | 33.2 |
| Bloom | 46.7 | 3.8 | 13.8 | 59.6 | 7.4 | 8.8 | 10.9 | 24.0 |
| Bloomz | 49.8 | 3.3 | 13.7 | 59.5 | 3.6 | 5.8 | 7.3 | 28.3 |
| Bloom-Rand | 52.9 | 3.9 | 15.4 | 61.0 | 7.3 | 13.2 | 14.8 | 30.1 |
| Bloom-Sbert | 57.5 | 4.3 | 16.5 | 61.9 | 9.9 | 14.0 | 16.2 | 33.1 |
| PlugMed | 61.0 | 4.1 | 19.3 | 62.8 | 8.2 | 13.0 | 14.6 | 37.3 |
| Pearson(%) | - | -22.7 | 48.9 | 59.7 | 27.1 | 42.8 | 41.2 | 71.3 |

Table 5: We evaluated three datasets by human ratings. For evaluation purposes, we randomly selected 100 samples from each dataset. The average score for each sample was calculated based on the assessments of three physicians.

sity, and Richness. This order also corresponds to the importance of the check items, and subsequent items only had test significance if the model satisfied the previous check item. Therefore, we apply the following rules for scoring. If a target response successfully pass a particular test, it receives a score of 20. However, if the target response does not pass a specific test, it is assigned a score of 0 for all subsequent check items.

We compute the average score of each model on the dataset and summarize the evaluation results in Table (5). Our analysis reveals that PlugMed exhibits the highest performance based on human evaluations. However, there still exists a significant gap between the responses generated by the models and the human responses, indicating that the models have not fully grasped the diagnostic capabilities. Additionally, we observe that BART performs well in the manual evaluation, primarily because we give a very low priority to richness. Appendix D provides a case study for illustrating this finding.

### 4.8 Analysis of Evaluation Metrics

Within this section, we have included an evaluation of the metric's reliability. To gauge the quality of these metrics, we employed the Pearson correlation coefficient to quantify their alignment with human evaluations. A score approaching 1 indicates a stronger metric performance. The corresponding outcomes can be found in Table (5).

Our examination indicates that the INT metric surpasses all others in performance, with Bert-Score coming in a close second. These two metrics exhibit stronger correlations with human assessment, indicating the superiority of a semantically based evaluation. Simultaneously, this outcome implies that the model's intent should align closely with the corpus. Lower scores for BLEU (Chen and Cherry, 2014) and higher scores for Rouge-L suggest a preference among individuals for more comprehensive model responses over precision-oriented ones. Lastly, based on the outcomes of T1M (exact match), T3M, and T5M, it becomes evident that factoring in term similarity is imperative when calculating term matching scores.

## 5 Related Work

### 5.1 Medical Dialogue Systems

According to system architecture, medical dialogue systems can be of two types: the pipeline and the end-to-end systems (Valizadeh and Parde, 2022). The pipeline systems usually involve four steps: the natural language understanding, the dialogue state

tracking, the dialogue action generation, and the natural language generation. Wei et al. (2018); Xia et al. (2020) propose to learn the dialogue policies for automated diagnosis using reinforcement learning. In other studies, Lin et al. (2019) proposes to model the associations between symptoms by constructing a symptom graph, aiming to enhance symptom diagnosis performance. Li et al. (2021a) proposes to use symptoms and diseases as keys to generate responses instead of dialogue states and actions based on a knowledge graph.

The end-to-end models, usually a sequence-to-sequence architecture (Sutskever et al., 2014), first attracted attention. The fine-tuning pre-trained models, such as GPT-2 (Radford et al., 2019), have been proven effective in task-based dialogue scenarios, as demonstrated by Su et al. (2021); Yang et al. (2021); Wang et al. (2022). BioBERT (Lee et al., 2020) and BioGPT (Luo et al., 2022) try to improve the performance by employing pre-training on medical corpora. Varshney et al. (2023b,a); Tang et al. (2023) propose explicitly using the Unified Medical Language System (UMLS) to incorporate knowledge in the dialogue generation process.

In summary, existing studies mainly emphasize the role of knowledge enhancement based on small models. In contrast, our work is conducted on knowledge-rich LLMs, emphasizing the enhancement of dialogue strategies.

### 5.2 ICL for Dialogue

As LLMs continue to advance, ICL has emerged as a new paradigm in natural language processing. The exploration of ICL's ability to evaluate and infer LLMs has become a prominent trend (Dong et al., 2022). Some studies have tried to apply ICL in dialogue generation. Roy et al. (2023) proposes a two-stage style transfer framework to leverage ICL for dialogue style transfer. Meade et al. (2023); Lee et al. (2022) employ a retrieval-based framework to mitigate bias and toxicity in chatbot-generated responses, guiding the model towards safer and more responsible dialogue. Hu et al. (2022); Chen et al. (2023) propose a method for long dialogue compression, enabling each prompt to contain more examples and improving dialogue state tracking performance. ICL has also been utilized for unsupervised generation of dialogue data in certain contexts (Li et al., 2022), expanding the potential applications of this approach.

Overall, the aforementioned works concentrate on either example retrieval or dialogue compression. In contrast, our work combines the two techniques for integrating multiple retrieval strategies.

## 6 Conclusion

In this paper, we use in-context learning to develop a patient-centered medical dialogue model. To this end, we introduce a Prompt Generation module capable of generating LLM input from both global and local views. Additionally, we construct a Response Ranking module using a supervised trained small model to filter the LLM output. Experimental results indicate that the responses generated by PlugMed exhibit a greater inclusion of comprehensive medical terms and PlugMed yields more accurate dialogue intents than other large language model based methods.

## Limitations

Based on human evaluation, we have identified shortcomings in PlugMed's diagnostic efficiency. This suggests that PlugMed has difficulty rapidly identifying the patient's disease, which may lead to an increase in average conversational discourse and harm the patient's experience. Our future work will prioritize addressing this issue.

## 7 Acknowledgement

Our work is supported by the National Key Research and Development Program of China (Project Number: 2020AAA0109400). We kindly appreciate all the researchers who provide valuable insights, discussions, and comments on this work.

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

# A  Details of Automatic Evaluation

## A.1  Intent Classifier

To compute Int, we employ the IMCS-IR dataset (Chen et al., 2022) and utilize Roberta-large (Liu et al., 2019) as the backbone model for training the classifier. To process a response, we concatenate it with the dialogue history and input the combined sequence into the model for classification. The input format is structured as follows:

 [dialogue history]  [response]

We extract the hidden layer vector of the '' token as the embedding representation for the response, which is subsequently classified using a two-layer neural network with a hidden dimension of 768. The intentions considered are presented in Table (6), and the model achieves an accuracy of **0.86** on the validation set.

| Action Type | Target |
|---|---|
| Request | Symptom |
| Request | Etiology |
| Request | Basic Information |
| Request | Existing Examination and Treatment |
| Inform | Drug Recommendation |
| Inform | Medical Advice |
| Inform | Precautions |
| Inform | Diagnose |
| Other | Other |

Table 6: The intents of the doctor.

## A.2  Term Extraction

Following the implementation outlined in (Li et al., 2021a), we employ word matching to extract high-frequency terms from responses. To accomplish this, we initially construct a glossary of medical high-frequency words. We then appropriately expand the glossary to ensure a satisfactory hit rate for term matching. Our approach is detailed as follows.

**Initialization of the glossary.** We utilized the Chinese medical high-frequency vocabulary acquired from THUOCL[6] as the initial glossary, comprising 18,749 frequently employed terms in doctor-patient communication.

**Glossary expansion.** We utilize the skip-gram algorithm to expand the list of high-frequency terms.

---

[6]https://github.com/thunlp/THUOCL

Our approach involves using the skip-gram algorithm to discover synonyms of the initial vocabulary and incorporate them into the vocabulary. This method helps us identify aliases and common names of the terms and facilitates the calculation of the TnM f1-score by providing term similarity information. To accomplish this, we train skip-gram word embeddings using the CMCQA (Weng, 2022) corpus, which consists of 1.3 million complete conversations, 19.83 million sentences, and 650 million tokens. Our skip-gram algorithm employs a word vector dimension of 300, a window size of 5, a sub-sample ratio of 3, and undergoes 8 rounds of training. Using the obtained word vectors, we enrich the high-frequency terminology list by adding the 10 synonyms with the closest semantic meanings for each term. Subsequently, we save the term vectors obtained from the training to facilitate the calculation of similarity scores.

**Discussion.** It is worth to note that Meddg (Liu et al., 2022) and a few other datasets propose similar evaluation metrics based on medical entity matching. This raises the question of why we are considering medical terminology matching instead of medical entity matching. We chose terminology matching instead of entity matching because we observe several issues.

Firstly, most medical named entity recognition (NER) datasets (Hongying et al., 2021; Guan et al., 2020) focus on recognizing a limited range of entity types, which often excludes entities crucial to the consultation process, such as etiology related to eating habits. Secondly, since our target audience is patients without medical expertise, the dialogue content incorporates numerous colloquial words. This poses challenges for NER models that are primarily designed for professional terminology. Consequently, the traditional NER model can only identify a limited number of entities within the dataset. For example, in the report of Meddg, an average sentence contains only 0.56 entities, which hampers the evaluation of the dialogue content.

Therefore, we believe that evaluating high-frequency medical terms would be a more logical approach. The glossary we selected offering an 81.9% coverage of the labeled entity classes in Meddg and featuring an average of 2.13 entities per utterance. Table (7) illustrates the comparison between Meddg's original entity extraction and our improved term extraction, clearly showing that our extracted terms better cover the dialogue content.

| Role | Dialogue History | Meddg | Ours |
|------|------------------|-------|------|
| Patient | How to treat colitis, the doctor prescribed reha-bilitation liquid and mesalachin granules, please read the bc sheet (Female, 35) | | |
| Doctor | What are your current symptoms? Have you had a gastroscopy? | Gastroscopy | Gastroscopy |
| Patient | I feel like I have diarrhea, but I can't get it out, and my stomach doesn't feel good, I don't know if it's serious, I don't have a gastroscopy. | | |
| Doctor | The medication prescribed by the doctor is symptomatic, just keep taking it. There may be ulcers in the intestines. Usually, you should also pay attention to your diet, avoid spicy stim-ulation, quit smoking and limit alcohol. | | Medicine, Intestinal, Ulcer, Diet |
| Patient | Is it necessary to do a gastroscopy? | | |
| Doctor | Because there is an ulcer in the intestine, the stomach should also be checked. But it is okay to take medicine first. | | Intestine, Ulcer, Stomach, Check |
| Patient | How long does this mesalachin need to take? | | |
| Doctor | The duration of medication is determined by the symptoms. When you feel well, you can stop the medication. Pay attention to your diet during the day. | | Symptoms, Duration of medication, Diet |

Table 7: A comparison of the entities provided by Meddg with the terms we extracted. It can be observed that our extracted terms can better cover the content of the conversation.

## A.3 Term Matching

Let $A_i = \{a_1^i, .., a_n^i\}$ represent the set of $n$ terms in the system response $i$, and $B_i = \{b_1^i, ..., b_m^i\}$ represent the set of $m$ terms in the standard response $i$. We use the cosine similarity of the skip-gram vector corresponding to the term as the required similarity score for Top-n Match. To calculate the f1-score, we define the following term types:

- Truth-Positive (TP): For any $a_u^i \in A_i$, if there exists $b_v^i \in B_i$ Top-n Matches with it, then we classify $a_u^i$ as a TP term.

- False-Positive (FP): If $a_u^i$ is not a TP term, it is classified as an FP term.

- False-Negative (FN): If $b_v^i$ does not top-n match with any $a_u^i \in A_i$, we classify $b_v^i$ as an FN term.

Then the calculation formula for micro-f1 is as follows, where N denotes the number of samples, and i denotes the index of the samples.

$$P = \frac{\sum_i^N TP_i}{\sum_i TP_i + \sum_i FP_i} \quad (3)$$

$$R = \frac{\sum_i^N TP_i}{\sum_i TP_i + \sum_i FN_i} \quad (4)$$

$$F = \frac{2PR}{P + R} \quad (5)$$

## B Implementation of Mars

The overview of the migration process are shown in Figure (5). Mars (Sun et al., 2022) incorporates two decoders, namely the Dialogue State Decoder and the Action State Decoder, which both utilize a Shared Encoder. The original workflow of Mars is as follows: 1) Utilize the Shared Encoder to encode the context $C$ and user input $U$, and employ the Dialogue State Decoder to decode the dialogue state $D$. 2) Employ the dialogue state to retrieve the corresponding entity $DB$ from the database. 3) Utilize Shared Encoder to encode $C$, $U$, and $DB$, and employ the Action State Decoder to decode the dialogue action $A$ and dialogue response $R$.

Since our task does not involve interaction with the database, we omit the second step but employ $D$ instead of $DB$. Considering that the dataset lacks labeled dialogue states and actions, we adopt the approach of Li et al. (2021a) and utilize high-frequency medical vocabulary found in sentences as the states and actions. Our subsequent experiments show that this approach facilitates the model to generate more medical terms while improving the fluency of the output. We keep the other settings the same as the original configuration.

## C ICL on Bloomz

This section explores the adaptability of Bloomz to In-Context Learning. We observed that the impact

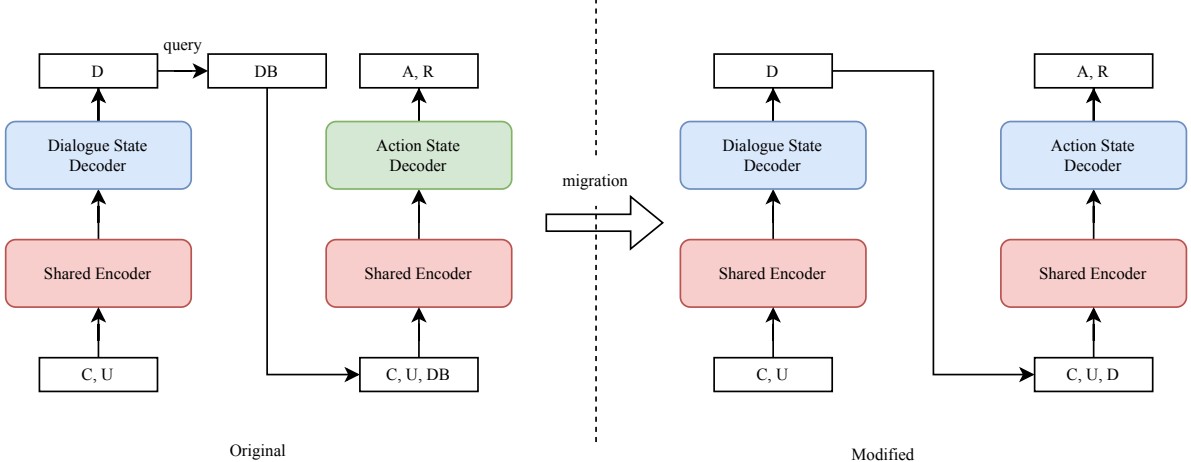

Figure 5: Migration methods for Mars models.

of ICL on BLoomz was minimal. Adjusting the order of the examples or changing the number of examples did not result in significant changes in the model's performance. To illustrate this phenomenon, we provide the details of the following experiment.

## C.1 Experimental Setup

In order to evaluate the impact of In-Context Learning (ICL) on the performance of Bloom and Bloomz, we conducted experiments on the validation set of the KaMed dataset. For each sample in the validation set, we generated 10 prompts, where each prompt consisted of a random number of randomly selected examples. The maximum number of examples included in a prompt was limited to less than 4. Using these 10 prompts, we made predictions using Bloom and Bloomz models respectively. We then count the number of non-repeats of the system's responses.

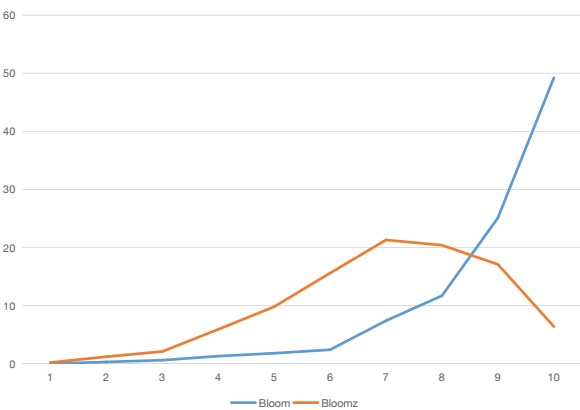

Figure 6: Diversity comparison of Bloom output and Bloomz output.

## C.2 Experimental Result

The test results are presented in Figure (6). The horizontal axis of the figure represents the number of unique outputs obtained from the 10 prompts per sample, while the vertical axis represents the percentage of samples in the dataset that have that specific number of unique outputs. From the figure, we can observe that Bloom generates almost different answers for ICL prompts containing different examples. In contrast, Bloomz generates more duplicate answers, indicating that Bloomz is less influenced by the sample. The observation that Bloomz's dialogue strategy is less malleable, suggests that the Instruct-tuning process may compromise the model's In-Context Learning capabilities on the medical dialogue generation task.

## D Case Study

We selected a representative sample from the KaMed dataset to facilitate an interpretable comparison of different models. The corresponding results are presented in Table (8). In the scenario depicted within this sample, the patient's mother provided limited information to the doctor, which is insufficient to support an accurate diagnosis. Consequently, the doctor made the decision to gather more necessary symptom information.

Our observations reveal that the Bart model exhibit an awareness of the information gap. However, Bart inquire about a known symptom, indicating that the small model lacks certain medical common sense and doesn't know what information to collect. Mars places greater emphasis on generating pertinent medical terms. In the absence of other

| History | U: The baby has a cold for the sixth day, bronchitis, phlegm, cough, and runny nose.
U: I went to the hospital of traditional Chinese medicine and prescribed Chinese medicine without decoction.
U: After taking it for three days, it didn't seem to work. What should I do (male, 13 months) |
|---|---|
| Golden | Hello! Does your baby have other symptoms? Such as fever? |
| BART | Hello, is your baby coughing badly? |
| Mars | The baby is bronchial sinusitis, bronchitis, cough, runny nose. |
| BLOOM | Your baby needs timely medical treatment. The child's current symptoms are obviously suppurative tonsillitis. In addition to taking anti-inflammatory treatment for this disease, it is recommended to perform tonsil removal surgery at the same time. |
| BLOOMZ | The baby is still young, so it is recommended to take western medicine. Chinese medicine has too many side effects on the baby, and the effect is not good. |
| ICL RAND | If your child coughs and has phlegm, you can take some phlegm-reducing medicines, such as Mucosolvan and keep warm to avoid catching cold. |
| ICL SBERT | The baby is now coughing and has phlegm. It is considered to be bronchitis. You can continue to take traditional Chinese medicine, or you can take cephalosporins, combined with atomization treatment. |
| PlugMed | Hello, does your baby have a fever now? |

Table 8: One case extracted from KaMed. Note that these dialogues are originally in Chinese, the English version may not get the same responses.

information, Mars choose to reiterate the patient's words in an attempt to enhance the likelihood of a terminological hit. Unfortunately, this approach result in the generation of responses of lower quality. The LLM produce lengthier responses, while ICL Rand and ICL Sbert attempt to provide a diagnosis directly. On the other hand, Bloom and Bloomz generate false and biased utterances, respectively, indicating a dearth of diagnostic strategies within these larger models. In contrast, PlugMed generate responses similar to the ground-turth, thus suggesting the effectiveness of our proposed method.

# E    Prompt Format

Our experiments employ two prompts: the Instruct prompt, shown in Figure (7), and the In-context prompt, shown in Figure (8). The Instruct prompt consists of a concise statement of the task goal, followed by direct input of all conversation histories into the model, serving as a test for the model's zero-shot capability. On the other hand, the In-context prompt involves presenting 4 examples prior to the samples to activate the model's imitation ability.

任务：扮演一名医生，继续写出以下医疗对话。
Task: Play as a doctor and continue writing the following medical dialogue.

患者: 胃炎胸闷，气短，消化不良（男，23岁）
Patient: Gastritis, chest tightness, shortness of breath, indigestion (Male, 23 years old)
医生: 您好，这种情况多久了，还有没有其他不适?
Doctor: Hello, how long has this been going on and is there any other discomfort?
患者: 两个多月了，打嗝，肚子咕咕的。浅表性胃炎这是功能性消化不良吗?
Patient: Over two months, hiccups, stomach rumbling. Is this functional indigestion with superficial gastritis?
医生: 不是，浅表性胃炎是浅表性胃炎，消化不良是功能性的诊断。你的情况从症状上看两种情况都有，有没有在吃什么药?
Doctor: No, superficial gastritis is superficial gastritis and indigestion is a functional diagnosis. From the symptoms you have both, are you taking any medication?
患者: 吃了些中药
Patient: I have taken some Chinese medicine.
医生: 西药吃了没有
Doctor: Have you taken any western medicine?
患者: 西药家里有伊托必利，雷贝拉唑，达喜。吃了一段时间吃了就舒服，停下就不舒服，打嗝很严重。请问我的体重怎么能恢复过来?
Patient: I have itopride, rabeprazole, and Daxil at home. I've been taking them for a while and I feel better when I eat them, but I don't feel better when I stop, and I have very bad hiccups. How can I get my weight back?
医生: 你看，你有的三种药还是要吃，建议连服4周。还有就是查一下幽门螺旋杆菌，如果幽门螺旋杆菌阳性，还要服药根治
Doctor: Look, you have three kinds of drugs still have to eat, it is recommended to take 4 weeks in a row. And you need to check the H. pylori, if the H. pylori infection is positive, you also need to take medicine to cure.
患者: 幽门螺杆菌阴性的这种胃病会导致消瘦吗为什么饭量上来体重恢复不过来
Patient: If the H. pylori infection is negative, can this stomach disease cause wasting and why I can't regain my weight when my appetite is back?
----
医生:
Doctor:

Figure 7: An example of the Instruct prompt.

对话摘要: 患者三个月前出现口腔扁桃体发炎，目前症状无明显改善。

Dialogue abstract: The patient developed inflammation of the oral tonsils three months ago, and her symptoms have not improved significantly.

患者：做胃镜检查一般多久需要检测一次啊？

Patient: How often do I usually need to be tested for gastroscopy?

医生：三个月吧，您有没有查过幽门螺旋杆菌?如果没有就三个月.

Doctor: Three months, I think. Have you ever checked for H. pylori, if not then three months.

患者：以前有吃了药现在没有了

Patient: I used to take medicine now I don't have it

----

医生：吃了14天的抗幽门螺旋杆菌的药物？[EOF]

Doctor: Have you taken enough anti-Helicobacter pylori medication for 14 days? [EOF]

对话摘要: 患者出现胃镜浅表性胃炎伴糜烂一周。

Dialogue abstract: Patient presented with gastroscopic superficial gastritis with erosion for one week.

医生：您查过幽门螺旋杆菌吗这个细菌一般会导致您的情况

Doctor: Have you been checked for H. pylori, a bacterium that generally causes your condition.

患者：单纯这个病严重吗该怎么治疗

Patient: I just want to simply ask if this disease is serious and how to treat it.

----

医生：既然您有症状一般需要杀菌耐信早晨一片口服一个月，胃三联口服一周就可以。然后所有药停药一个月后，复查幽门螺旋杆菌[EOF]

Doctor: Since you have symptoms generally need to sterilization resistance letter in the morning a piece of oral one month, gastric triple oral one week on. Then all drugs stop a month after the review of H. pylori [EOF]

对话摘要: 患者三周前出现胃炎伴糜烂症状，口服雷贝拉唑治疗三周。

Dialogue abstract: The patient developed gastritis with erosion three weeks ago and was treated with Rabeprazole for three weeks.

患者：那药店里叫我一天吃四粒，隔天再吃

Patient: The pharmacy told me to take four capsules a day, and then take them every other day.

医生：可以啊

Doctor: That's fine.

患者：以前是胀的，现在雷贝拉唑吃了三周，感觉还好，就喉咙有异物感

Patient: I've been taking rabeprazole for three weeks and I feel fine, just a foreign body feeling in my throat.

医生：嗯这个慢慢也会消除的

Doctor: Well, this will slowly be eliminated

患者：那你意思喉咙有异物感正常不用食管镜吗

Patient: Do you mean I don't need an esophagoscope for a foreign body feeling in the throat?

----

医生：不用的 [EOF]

Doctor: Not necessary [EOF]

对话摘要: 患者两月前出现腹胀症状，目前症状无明显改善。

Dialogue abstract: The patient developed bloating two months ago and currently has no significant improvement in symptoms.

患者：没有，我目前吃了幽门四联药，然后是益生菌和四磨汤。大便不成形，特别容易腹泻

Patient: No, I am currently taking pyloric quadruple medication followed by probiotics and tetracycline. Currently the stools are not formed and I am particularly prone to diarrhea.

医生：吃这两个药，同时配合吃点益生菌就行

Doctor: Eat these two drugs, and at the same time take some probiotics.

患者：有没有精神治疗的药物？

Patient: Is there any psychiatric medication?

----

医生：那可以吃点黛力新但需要去医院专科开[EOF]

Doctor: You can eat some flupentixol but need to go to the hospital specialist to prescribe [EOF]

对话摘要:患者两月前出现打嗝，胸闷，气短，目前症状无明显改善。

Dialogue abstract:The patient developed hiccups, chest tightness, and shortness of breath two months ago, and her current symptoms have not improved significantly.

医生: 你看，你有的三种药还是要吃，建议连服4周。还有就是查一下幽门螺旋杆菌，如果幽门螺旋杆菌阳性，还要服药根治

Doctor: Look, you have three kinds of drugs still have to eat, it is recommended to take 4 weeks in a row. And you need to check the H. pylori, if the H. pylori infection is positive, you also need to take medicine to cure.

患者: 幽门螺杆菌阴性的这种胃病会导致消瘦吗为什么饭量上来体重恢复不过来

Patient: If the H. pylori infection is negative, can this stomach disease cause wasting and why I can't regain my weight when my appetite is back?

----

医生:

Doctor:

Figure 8: An example of In-context prompt. Blue text indicates examples and red text indicates the sample.