# OpenReview forum: "PlugMed: Improving Specificity in Patient-Centered Medical Dialogue Generation using In-Context Learning"
_EMNLP/2023/Conference — EMNLP 2023 Findings_

### Official Review · Reviewer_X2S1 · 2023-07-24

**Soundness:** 4

**Excitement:**

4: Strong: This paper deepens the understanding of some phenomenon or lowers the barriers to an existing research direction.

**Paper Topic And Main Contributions:**

In this paper, the authors propose a patient-centred medical dialogue model based on prompt learning from large language model (LLM) called PlugMed, which is equipped with a prompt generation (PG) module and a response ranking (RR) module to enhance LLMs’ dialogue strategies for improving the specificity of the responses.
Experimental results indicate that the responses generated by PlugMed exhibit a greater inclusion of comprehensive medical terms and PlugMed yields more accurate dialogue intents than other large language model-based methods.

**Reasons To Accept:**

1. The authors a patient-centred medical dialogue model called PlugMed, which enhances LLMs to generate responses that conform to the diagnostic strategy.
2. The authors propose two evaluation metrics, i.e., the intent accuracy and the high-frequency medical term accuracy, where the intent accuracy is used to evaluate the reasonableness of the dialogue actions adopted by the system, and the high-frequency medical term accuracy focuses on measuring the presence of essential medical information in the system’s responses.
3. Experimental results indicate that the responses generated by PlugMed exhibit a greater inclusion of comprehensive medical terms and PlugMed yields more accurate dialogue intents than other large language model-based methods.

**Reasons To Reject:**

The author only compares their method with 4 LLMs, and these LLMS are not state-of-the-art.

**Reproducibility:**

5: Could easily reproduce the results.

**Reviewer Confidence:**

4: Quite sure. I tried to check the important points carefully. It's unlikely, though conceivable, that I missed something that should affect my ratings.

---

> ### Author Rebuttal · Authors · 2023-08-24
>
> **The author only compares their method with 4 LLMs, and these LLMS are not state-of-the-art.**
>
> We sincerely appreciate the reviewer's insightful feedback. We opted to utilize the Bloom family primarily due to its complete open-source nature. This choice stems from our conviction that conducting experiments on a fully open-source LLM yields enhanced interpretability. To assess the efficacy of our approach, we will complement the final version with experiments involving Chatgpt, thus confirming its generalizability.

---

### Official Review · Reviewer_V8yn · 2023-08-04

**Typos Grammar Style And Presentation Improvements:** NA
**Soundness:** 3

**Excitement:**

3: Ambivalent: It has merits (e.g., it reports state-of-the-art results, the idea is nice), but there are key weaknesses (e.g., it describes incremental work), and it can significantly benefit from another round of revision. However, I won't object to accepting it if my co-reviewers champion it.

**Missing References:**

NA

**Paper Topic And Main Contributions:**

This paper presents a new approach to dialogue response generation in medical consulting scenarios.

The primary purpose is to enhance the pre-trained general-purpose Chat-LLM with in-context learning and response re-ranking methods.
Moreover, two new metric scores for medical dialogue generation are proposed by measuring 1) dialogue intent and 2) specific medical terms.

**Questions For The Authors:**

1. What data is the SLM trained on? The dialogue corpus in the selected datasets? If so, it makes sense that it will select the response more similar to the datasets.

2. The baseline model which is simply using SBERT and history to perform retrieval. What is the different with Global retrieval? And the performance simply with SBERT is very competitive and even outperforms all other methods in some cases. Then, what is the necessity for global retrieval and local retrieval?

3. The new evaluation metric requires a set of medical terms. How the terms are collected? The quality of the terms. Does it has a wide coverage of different medical domains?

**Reasons To Accept:**

1. The paper is in general well-written and easy to follow.

2. The proposed method is logically sound. Some evidence is given to validate the proposed ideas.

3. The paper proposes a couple of things including 1) a retrieval method, 2) a reranking method 3) two evaluation metrics specifically for medical dialogue generation.

**Reasons To Reject:**

1. In this paper, only BLOOM model is used as the backbone. The paper claims the proposed method is a plug-and-play module. Therefore, the effect on multiple backbone LLMs should be shown to prove its generalizability and the effect also applies to other models. Also, it is shown ICL does not help with the BLOOMZ model, which also demands further investigation on other LLMs.

2. The re-ranking approach is somehow heuristic to me. The BART model is producing more chitchat outputs with lower information density. Using the perplexity of BART to rank LLM outputs will result in choosing output with less information, which is claimed to be more human-like. All validations and analyses are with BLOOM(Z) model which is shown to easily give conclusions. However, it is not clear how another model performs. Besides, from my experience, changing the prompts can also guide the model to be more careful before making medical conclusions/suggestions.

3. Overhead computation and grounding knowledge. The plug-in-and-play model introduces additional overhead for retrieval and reranking. A discussion on this is needed. Meantime, it requires a medical conversation pool to retrieve the samples. Currently, the setting is to retrieve samples from the same dataset. It may lead to bias because similar dialogue styles and domains are covered. The generalizability to other data sources is not very clear.

4. I have some questions which are not clearly explained in the current manuscript (or I missed them) and are elaborated on 'Question for the authors'.

**Reproducibility:**

4: Could mostly reproduce the results, but there may be some variation because of sample variance or minor variations in their interpretation of the protocol or method.

**Reviewer Confidence:**

4: Quite sure. I tried to check the important points carefully. It's unlikely, though conceivable, that I missed something that should affect my ratings.

---

> ### Author Rebuttal · Authors · 2023-08-24
>
> **In this paper, only BLOOM model is used as the backbone. The paper claims the proposed method is a plug-and-play module. Therefore, the effect on multiple backbone LLMs should be shown to prove its generalizability and the effect also applies to other models. Also, it is shown ICL does not help with the BLOOMZ model, which also demands further investigation on other LLMs.**
>
> Thank you for your suggestion. We primarily utilize Bloom due to its dual merits of being open source and interpretable. These qualities empower us to formulate the ensuing conjectures. We think that our approach possesses theoretical applicability to the majority of the LLM's foundational models—those that have exclusively undergone unsupervised training. However, problems may arise if our approach is applied to models that have undergone instruct tuning or RLHF.
>
> We posit that the root cause for this phenomenon could be attributed to the nature of the training tasks to which the model has been exposed. For instance, Bloomz's instruction fine-tuning dataset, designated as bigscience/xP3, predominantly comprises single-round QA tasks, lacking the inclusion of multi-round dialog tasks. It is conceivable that this pre-training emphasis on single-round QA potentially compromised the model's adeptness in partaking in multi-round dialogues. Simultaneously, the dataset primarily orients itself towards the general domain, potentially hampering the model's prowess in specialized domains. Our experimental findings demonstrate that such issues were not encountered in the case of ChatGPT.
> We randomly selected 1000 samples from the Meddg dataset to test the ChatGPT, and the results are as follows:
>
> | **Model**        | **BLEU** | **T1M@P** | **T1M@R** | **T1M@F** | **T3M@P** | **T3M@R** | **T3M@F** | **T5M@P** | **T5M@R** | **T5M@F** | **INT** |
> | ---------------- | -------- | --------- | --------- | --------- | --------- | --------- | --------- | --------- | --------- | --------- | ------- |
> | Vanilla          | 4E-05 | 3.5       | 22.8      | 6.1       | 6.4       | 39.1      | 11        | 8.3       | 48.7      | 14.2      | 23      |
> | Global View      | 0.4      | 5.0       | 15.9      | 7.6       | 8.3       | 26.0      | 12.6      | 10.2      | 32.0      | 15.5      | 32.2    |
> | Local Primary    | 0.4      | 4.8       | 15        | 7.3       | 8.4       | 25.6      | 12.6      | 10.1      | 31.1      | 15.3      | 29.1    |
> | Local Secondary  | 0.4      | 4.7       | 15.1      | 7.2       | 8.1       | 25.3      | 12.3      | 10.1      | 31.3      | 15.3      | 31.5    |
> | \+Ranking (Ours) | 0.4      | 5.1       | 16.4      | 7.7       | 8.7       | 27.4      | 13.2      | 10.8      | 33.3      | 16.3      | 33.6    |
>
>
> We observed that, ICL played a pivotal role in enhancing ChatGPT's diagnostic capabilities. At the same time, our rerank strategy is also valid for the model. Meanwhile, a noticeable discrepancy arises in the comparison of ChatGPT's BLEU and INT scores with those of Bloom's. Analysis of the generated output reveals a discernible pattern wherein ChatGPT frequently refrains from directly addressing the patient's inquiries. Instead, it frequently opts to conclude the diagnostic procedure by advising the patient to seek medical consultation. This strategy avoids making mistakes but results in a much less useful model. Consequently, one could reasonably posit that the integration of security protocols into ChatGPT by OpenAI might have compromised the model's diagnostic capabilities.
>
> Due to the additional time required to supplement the human evaluations, we will add the full experimental content of ChatGPT in the revised version.
>
> **The re-ranking approach is somehow heuristic to me. The BART model is producing more chitchat outputs with lower information density. Using the perplexity of BART to rank LLM outputs will result in choosing output with less information, which is claimed to be more human-like. All validations and analyses are with BLOOM(Z) model which is shown to easily give conclusions. However, it is not clear how another model performs. Besides, from my experience, changing the prompts can also guide the model to be more careful before making medical conclusions/suggestions.**
>
> Thanks for your comments. The rerank model functions mainly based on whether the dialog actions are reasonable or not, but not only on the information density. That is why the INT and TnM metrics be improved after reordering. There is no necessary causal relationship between information density and whether the dialog action is correct or not. At the same time a decrease in information density is likely to lead to a decrease in TnM@R, i.e., the rerank model selects replies that contain fewer medical entities. From the results of the ablation experiments, PlugMed's TnM@R is not the lowest among the four strategies, and on some datasets, instead, it has a high ranking.
>
> We observed that Bloom and Bart diverge the most in their choice of whether to take the "inform" action. Therefore, instead of interpreting this phenomenon in terms of information density, we propose a more plausible explanation: the rerank model, with its conservative nature (tends to request), would penalize overconfident responses (tends to inform), thereby acting as a counterbalance. Due to the dialogue action are corrected, the hit rate for medical entities is similarly improved.
>
> For the last question, we believe that enhancing the model's self-awareness by only imposing instructions may be overly simplistic. The model must not only act with caution, but also understand the relevant data collection priorities that depend on different diseases. The dynamic nature of collected information on various diseases makes the formulation of precise instructions complex. This is the role of ICL in alleviating complex instruction challenges.
>
> **Overhead computation and grounding knowledge. The plug-in-and-play model introduces additional overhead for retrieval and reranking. A discussion on this is needed. Meantime, it requires a medical conversation pool to retrieve the samples. Currently, the setting is to retrieve samples from the same dataset. It may lead to bias because similar dialogue styles and domains are covered. The generalizability to other data sources is not very clear.**
>
> Thank you for your valuable feedback and insightful questions. The primary overhead lies in the generation of multiple responses, a process that frequently exceeds ten seconds (LLMs are slow). This is in contrast to the minimal time required for retrieval and reranking. We will add the description of this in the revised version. Moreover, the dataset we have selected originates from China's widely used online medical consultation platform. The platform has registered doctors from all over the country covering the most common types of diseases. As a result, we hold the conviction that this dataset adequately encompasses a diverse array of situations within medical consultation scenarios.
>
> **What data is the SLM trained on? The dialogue corpus in the selected datasets? If so, it makes sense that it will select the response more similar to the datasets.**
>
> The SLM undergoes training using the dialog dataset employed by ICL. Subsequently, it possesses the capability to rectify the diagnostic strategy of the LLM. It is necessary to highlight that the SLM doesn't directly interfere with the LLM's response generation process. As a result, the ultimately chosen responses retain their comprehensiveness and depth of medical knowledge.
>
> **The baseline model which is simply using SBERT and history to perform retrieval. What is the different with Global retrieval? And the performance simply with SBERT is very competitive and even outperforms all other methods in some cases. Then, what is the necessity for global retrieval and local retrieval?**
>
> SBERT and Global retrieval are indeed same method. Our research revealed that Global retrieval's efficacy is influenced by the length of conversation history, leading to a decline in retrieval quality as the history lengthens. In response to this challenge, we introduced Local retrieval. This variant remains unaffected by conversation length, bolstering its performance with lengthy conversations. Substantiation for this lies in the comparable performance of Global retrieval and PlugMed on MedDialog, characterized by an average of 4.76 conversation rounds. Conversely, on MedDG (averaging 9.92 rounds) and KaMed (averaging 11.62 rounds), PlugMed notably outperforms Global retrieval.
>
> Meanwhile, we believe that the effective performance of Global Retrieval originates from the long-tailed distribution of sample lengths. In particular, during testing, the dialog session is commonly divided into multiple samples based on the dialog rounds. Clearly, a majority of these samples encompass only brief conversational histories. Such a testing approach significantly in favor of Global Retrieval strategy. Nonetheless, due to the frequent occurrence of lengthy conversations in real-world situations, integrating Local Retrieval becomes essential.
>
> We will add the discussion of this issue in the revised version.
>
> **The new evaluation metric requires a set of medical terms. How the terms are collected? The quality of the terms. Does it has a wide coverage of different medical domains?**
>
> We appreciate your insightful questions regarding our evaluation metric. The collection of medical terms for our metric is facilitated through THUOCL (THU Open Chinese Lexicon), a high-quality Chinese thesaurus compiled by Tsinghua University's Laboratory of Natural Language Processing and Social and Humanities Computing. This compilation incorporates terms from diverse sources, including mainstream websites, hot keywords, and input methods thesaurus. Rigorous manual screening has been undertaken in multiple iterations to ensure the precision of included terms. Upon examination, we found that the thesaurus demonstrates a commendable breadth by encompassing various medical domains, including diseases, symptoms, drugs, and food. Its contents adhere to higher standards of terminological uniformity and reliability. We will add the explanation of this issue in the revised version.

---

### Official Review · Reviewer_nsEF · 2023-08-05

**Soundness:** 3

**Ethical Concerns:**

Yes

**Excitement:**

3: Ambivalent: It has merits (e.g., it reports state-of-the-art results, the idea is nice), but there are key weaknesses (e.g., it describes incremental work), and it can significantly benefit from another round of revision. However, I won't object to accepting it if my co-reviewers champion it.

**Justification For Ethical Concerns:**

The paper overlooks an essential aspect of ethical considerations, especially in the context of medical terminology. An ethical discussion is crucial when dealing with medical data and applications.

**Paper Topic And Main Contributions:**

The paper introduces a two-module framework for generating responses and employs a two-step evaluation metric. The primary objective is to achieve automatic Medical Dialogue response generation using Language Models (LLMs). This is accomplished by harnessing In-context learning and a two-component system comprising a prompt generation module and a response ranking module.

**Questions For The Authors:**

> The authors did not report standard evaluation metrics for dialogue generation, such as ROUGE, METEOR, and PPL. Additionally, there is a lack of semantic evaluation metrics, such as BERT Score, which could have provided further insights.

> The paper lacks a discussion on the scale of models used. Failing to demonstrate significant efficacy with only a small gain in performance, especially considering the increased complexity of the models, raises concerns.

**Reasons To Accept:**

1. The authors take into consideration both the global context and the local context for generating responses. The global context captures similar dialogues from the dataset, and the local context is used to extract symptom information for generating medically appropriate responses.

2. The authors test their model on three datasets containing a substantial number of dialogues which gives assurance to the credibility of the proposed method.

3. The intent accuracy metric guarantees reasonable response generation, and the high-frequency medical term accuracy makes sure appropriate medical terms and information is present in the generated responses.

**Reasons To Reject:**

1. The paper lacks technical novelty as most of its components are well-known, simply combined into a system. Moreover, it fails to address the technical challenges relevant to the research.

2. Detailed illustrations or elaborate descriptions of the internal structure of the prompt generator, SLM Encoder, and SLM Decoder are missing in the paper. Although the mentioned frameworks are noted, the specific architectures remain inadequately explained.

3. The authors did not report standard evaluation metrics for dialogue generation, such as ROUGE, METEOR, and PPL. Additionally, there is a lack of semantic evaluation metrics, such as BERT Score, which could have provided further insights.

4. The paper lacks a discussion on the scale of models used. Failing to demonstrate significant efficacy with only a small gain in performance, especially considering the increased complexity of the models, raises concerns.

5. The paper overlooks an essential aspect of ethical considerations, especially in the context of medical terminology. An ethical discussion is crucial when dealing with medical data and applications.

**Reproducibility:**

3: Could reproduce the results with some difficulty. The settings of parameters are underspecified or subjectively determined; the training/evaluation data are not widely available.

**Reviewer Confidence:**

4: Quite sure. I tried to check the important points carefully. It's unlikely, though conceivable, that I missed something that should affect my ratings.

---

> ### Author Rebuttal · Authors · 2023-08-24
>
> **The paper lacks technical novelty as most of its components are well-known, simply combined into a system. Moreover, it fails to address the technical challenges relevant to the research.**
>
> We appreciate your feedback and concerns. We think that the concern "lack of technological innovation" in our paper may raise from a paradigm shift in the research approach, which we'd like to elucidate. Owing to the large size of LLMs and the high cost of their training, the application of Prompts Engineering techniques, including ICL and CoT, are frequently employed for the adaption of downstream tasks. Prompts Engineering is a training-free methodology that strives to formulate prompt generation strategies, and it is an important means of realizing Artificial General Intelligence (AGI). This kind of research does not emphasis on the model design but treats the off-the-shelf AI models as callable services. **The challenge with prompt engineering lies in the design of a task-oriented prompt generation strategy**. We have experimentally demonstrated that our proposed approach is effective in improving the dialog strategy of LLM and solve the problems of example retrieval as well as example compression in ICL tasks. Therefore, we believe we have addressed the technical challenges in our research.
>
> Your feedback highlights the need for a clearer distinction between ICL and traditional methods in the relevant work section. We will remedy this deficiency in the final version of the paper.
>
>
> **Detailed illustrations or elaborate descriptions of the internal structure of the prompt generator, SLM Encoder, and SLM Decoder are missing in the paper. Although the mentioned frameworks are noted, the specific architectures remain inadequately explained.**
>
> Thank you for the advice. At present, the detailed information about encoders and decoders have not been included, as we feel that the module implementation is mainly related to the training data, not its internal structure. And training with a conventional encoder-decoder pre-trained model can get basically the same effect. The encoder-decoder pre-trained model we adopted in this paper is BART model. We will add this discussion in the revised version.
>
> **The authors did not report standard evaluation metrics for dialogue generation, such as ROUGE, METEOR, and PPL. Additionally, there is a lack of semantic evaluation metrics, such as BERT Score, which could have provided further insights.**
>
> We understand the reviewer's concerns about missing some standard evaluation metrics. The reason why we did not include those metrics stems from the specific nature of our task, which is to complete the Task-Oriented Dialogue (TOD). In this case for evaluating the dialog quality, Success Rate, Inform Rate, and BLEU are widely used. These metrics can measure the accuracy of dialogue actions and entities mentioned in dialogs, which is more in line with the essence of our evaluation criteria. Of course, we can include some of the experiment results for those missed metrics in the revised version if it is necessary.
>
> Another pertinent aspect is that LLM typically generates a more comprehensive response. However, in real-world situations, medical practitioners usually choose concise answers. This tendency puts LLM at a relative disadvantage in these standard evaluation metrics. Nevertheless, it is worth emphasizing that a more comprehensive response always helps to enrich the patient experience. We believe that this is an advantage of LLM, but these standard evaluation metrics may unintentionally underestimate this advantage.
>
> **The paper lacks a discussion on the scale of models used. Failing to demonstrate significant efficacy with only a small gain in performance, especially considering the increased complexity of the models, raises concerns.**
>
> Thank you for your thoughtful comment. Indeed, we should report the scale of the model used here. We will include the information in the revised version. Of course, in the work, our research goal is to explore how to use non training methods to enable LLM to perform medical dialogue tasks. Because we do not need to train the LLM, the number of model parameters is not the focus of our consideration, and we do not affect the generalization ability of LLM. Meanwhile, we have experimentally demonstrated that the generalized model outperforms the specialized model in specific fields. Our other contribution is to demonstrate the effectiveness of the research path of “large model + strategy injection”, which is different from the traditional method of “small model + knowledge injection”.
>
> **The paper overlooks an essential aspect of ethical considerations, especially in the context of medical terminology. An ethical discussion is crucial when dealing with medical data and applications.**
>
> Thank you for your comment. We appreciate your emphasis on the importance of ethical considerations, particularly within the realm of medical terminology and data. We will enhance the paper by incorporating a comprehensive discussion on ethical implications, underscoring the significance of responsible handling and applications of medical data.

---

### Meta-Review · Area_Chair_oics · 2023-09-11

**Recommendation:** 3

**Metareview:**

In this paper, the authors propose a patient-centered medical dialogue model named PlugMed, which leverages prompt learning from large language models (LLM). PlugMed is equipped with two key modules: a prompt generation (PG) module and a response ranking (RR) module. These components work in tandem to enhance LLMs' dialogue strategies, with the primary goal of improving response specificity.
The experimental results presented in the paper demonstrate that PlugMed outperforms other large language model-based methods by achieving more accurate dialogue intents.

However, I agree with one of the reviewers that this paper primarily combines established components into a unified system and lacks novelty. Another shortcoming of the paper is the absence of standard evaluation metrics commonly used for dialogue generation tasks, such as ROUGE or BLEU. These metrics are essential for assessing response quality and should be included to strengthen the paper's empirical findings.

---

### Decision · Program_Chairs · 2023-10-07

**Decision:**

Accept-Findings

**Comment:**

In this paper, the authors propose a patient-centered medical dialogue model named PlugMed, which leverages prompt learning from large language models (LLM). PlugMed is equipped with two key modules: a prompt generation (PG) module and a response ranking (RR) module. These components work in tandem to enhance LLMs' dialogue strategies, with the primary goal of improving response specificity.
The experimental results presented in the paper demonstrate that PlugMed outperforms other large language model-based methods by achieving more accurate dialogue intents.

However, I agree with one of the reviewers that this paper primarily combines established components into a unified system and lacks novelty. Another shortcoming of the paper is the absence of standard evaluation metrics commonly used for dialogue generation tasks, such as ROUGE or BLEU. These metrics are essential for assessing response quality and should be included to strengthen the paper's empirical findings.